# Recombination in Bacterial Genomes: Evolutionary Trends

**DOI:** 10.3390/toxins15090568

**Published:** 2023-09-12

**Authors:** Anton E. Shikov, Iuliia A. Savina, Anton A. Nizhnikov, Kirill S. Antonets

**Affiliations:** 1Laboratory for Proteomics of Supra-Organismal Systems, All-Russia Research Institute for Agricultural Microbiology (ARRIAM), 196608 St. Petersburg, Russia; a.shikov@arriam.ru (A.E.S.); iu.savina@arriam.ru (I.A.S.); a.nizhnikov@arriam.ru (A.A.N.); 2Faculty of Biology, St. Petersburg State University (SPbSU), 199034 St. Petersburg, Russia

**Keywords:** homologous recombination, horizontal gene transfer, pathogenesis, symbiosis, ecological adaptation, HR, HGT

## Abstract

Bacterial organisms have undergone homologous recombination (HR) and horizontal gene transfer (HGT) multiple times during their history. These processes could increase fitness to new environments, cause specialization, the emergence of new species, and changes in virulence. Therefore, comprehensive knowledge of the impact and intensity of genetic exchanges and the location of recombination hotspots on the genome is necessary for understanding the dynamics of adaptation to various conditions. To this end, we aimed to characterize the functional impact and genomic context of computationally detected recombination events by analyzing genomic studies of any bacterial species, for which events have been detected in the last 30 years. Genomic loci where the transfer of DNA was detected pertained to mobile genetic elements (MGEs) housing genes that code for proteins engaged in distinct cellular processes, such as secretion systems, toxins, infection effectors, biosynthesis enzymes, etc. We found that all inferences fall into three main lifestyle categories, namely, ecological diversification, pathogenesis, and symbiosis. The latter primarily exhibits ancestral events, thus, possibly indicating that adaptation appears to be governed by similar recombination-dependent mechanisms.

## 1. Introduction

Through shaping the genomic landscape, horizontal gene transfer (HGT) or lateral gene transfer (LGT) and homologous recombination (HR) both serve as principal evolutionary forces in bacteria. These mechanisms provide genetic plasticity and thereby ensure adaptation to ecological niches [1], regulate virulence [2] and increase fitness [3]. The effect of these events is not only constrained to bacteria, but also plays a vital role in orchestrating the evolution and adaptability of archaea, viruses, and even eukaryotes [4]. The first phenomenon mentioned implies the replacement of DNA sequences when affecting genomic loci with contiguous, highly homologous regions [5]. HGT, in its turn, could be roughly defined as the acquisition of genetic material from a donor to recipient bacterial cells, predominantly requiring micro-homology sufficient for the incorporation of exogenous DNA into bacterial chromosomes or plasmids [6,7]. Not only do HR and HGT entail speciation [8] and spark the origin of new strains [4], but also cause antibiotic resistance [9], enhanced virulence [10], and reduced efficiency of vaccines [11]. However, the outcomes of these processes, in some cases, are beneficial for industries insofar as they result in the modulation of symbiotic relationships with agriculturally important plants [12] or lead to the emergence of strains capable of metabolizing pollutants, thus exhibiting promising biotechnological potential [13].

In terms of the parts of the bacterial genome affected, HR was reported to exert an effect on core genes [14], i.e., those shared by the majority of isolates within a certain population, whereas the accessory component is commonly embedded into genomic regions via HGT [15]. Therefore, the former alters allelic diversity, and the latter modifies gene composition. The preliminary step preceding the import or insertion of loci is DNA acquisition. Foreign DNA could enter bacterial cells via three fundamental mechanisms, namely, transformation, transduction, and conjugation [16]. Transformation involves a direct influx of DNA from the environment and is observed in diverse pathogens, including the genera *Neisseria*, *Helicobacter*, *Streptococcus*, etc. [7]. Being a complex multi-stage procedure, it is comprised of multiple steps, such as the activation of competence in the early stages of bacterial growth, triggering a two-component system ComD/ComE, and the action of the type IV pilus apparatus, through which foreign DNA transfer is performed [7]. The import is accompanied by DNA processing into 6 kb fragments by surface endonuclease [7]. Transduction is a phage-mediated DNA transfer between cells, whereby the phage carrying genetic material can integrate into a host genome [7]. The embedded loci could represent virulence determinants, e.g., toxin-encoding genes [17]. Conjugation, first discovered in *E. coli* when characterizing self-replicating F-plasmids, is carried out through direct cell-to-cell contact [16]. This necessity for maintaining cell contacts does not allow the transmission of genetic material other than small genetic elements, e.g., plasmids. It may also regulate the transfer of integrative and conjugative elements (ICEs), which lack a self-replication system and are capable of copying only during conjugation [7,16].

Upon obtaining DNA, either homologous or site-specific recombination occurs, with the latter governing HGT. HR is accompanied by the formation and consequent resolution of the Holliday junction [18]. It requires the RecA protein with polymerase activity attached to single-stranded DNA stabilized with SSB (single-stranded DNA-binding protein) [18]. Two alternative protein complexes, RecBCD and RecFOR, participate in the downstream stages [7,19]. Noteworthy, bacterial genomes demonstrate a non-uniform distribution of recombination systems with the prevalence of RecFOR over RecBCD in some organisms, presumably indicating the redundancy of the molecular mechanisms governing recombination [19]. Alternatively, obligate endosymbionts lack the RecA protein, while in their genomes, RecA-independent recombination has been reported [19,20,21], which can probably be explained by tandem repeats inducing genetic exchange [22]. HGT, understood as the incorporation of mobile genetic elements (MGEs) including plasmids, prophages, ICEs, and pathogenicity islands, is mainly mediated by site-specific recombination [7]. The respective integration is carried out by tyrosine recombinases, also known as integrases, and serine recombinases called resolvases [18]. The first family comprises XerC and XerD proteins, which separate chromosomal dimers during bacterial replication occurring at the *dif* site [18], whereas serine recombinases operate by making four breaks in double-stranded DNA molecules with their subsequent ligation [7].

HR and HGT are tightly interconnected, given that horizontally acquired genes are often flanked with genomic regions with an excessive HR frequency. This possibly serves to regulate genome size through excising obtained genes [23,24]. Additionally, transmitted MGEs could further engage in HR, as shown for prophages in cases of co-infection [25]. Moreover, pathogenic islands, insertion sequences, and other imported loci are characterized by HR signals predicted through bioinformatics tools [9,26,27,28]. On this account, in the current review, we would not distinguish between site-specific recombination-driven HGT and HR, using an umbrella term recombination instead.

Comparative studies have examined the intensity of HGT and HR to be unequal within bacterial populations that belong to different groups according to the ecological niches they occupy. As an example, obligate pathogens and symbionts are characterized by their lower HR rates than free-living organisms, commensals, and opportunistic pathogens [29,30]. Similar results held when considering horizontally transferred genes being presented by infection effectors and antibiotic resistance factors, thus, serving as modulators of bacterial invasion and promoting adaptation to certain hosts [3]. Taking into consideration the above-mentioned information, an in-depth understanding of the functional effect that recombination exerts on bacterial populations is needed concerning fundamental science and practical implications. In the review presented, we analyze aspects of recombination by discussing studies made in the last 30 years. The included studies comprised 91 bacterial species from different taxonomic groups. When choosing the articles, we focused on those in which recombination signals were found through computational predictions in whole genomes and/or individual loci, contiguous regions, and extrachromosomal elements. We summarize which parts of the genome are subjected to recombination and provide a scheme illustrating the biological roles of proteins encoded by the respective chimeric or acquired loci (Figure 1). We reveal that the vast majority of observations (Appendix A) fall into three categories related to three extremely dynamic processes: the establishment and development of (i) symbiotic or (ii) pathogenic relationships and (iii) ecological diversification caused by alterations in the environment.

## 2. Functional Impact of Recombination

### 2.1. Ecological Adaptation

Here, ecological adaptation and diversification mean the successful transfer to different ecotopes, or even new niches accompanied by genomic diversification leading to specification. Recombination tends to occur when bacterial populations occupy geographically isolated regions, as shown for *Alteromonas macleodii* at different depths [1], *Bacillus cereus*/*B. thuringiensis* isolates in Norway [31], *Arthrobacter* sp. in glaciers [32], the freshwater bacterium *Polynucleobacter asymbioticus* [33], and *Thermotoga maritima* subpopulations in oil reservoirs [34]. Competitiveness is mainly maintained by either obtaining metabolic pathways for catabolizing new energy sources or maintaining the diversity of synthesized toxic compounds. For example, recombination allowed *Pseudomonas stutzeri* [13] and *Arthrobacter nicotinovorans* [35] to metabolize naphthalene and nicotine, respectively. Another recombinational scenario leads to strain diversity based on the synthesized toxins, so-called chemotypes, namely, nonribosomal toxic peptides of *Planktothrix* sp. [36] and *Microcystis* sp. [37] or the diverse antimicrobial secondary metabolites of *Streptomyces* sp. [38].

### 2.2. Symbiotic Relationships

The genomic evolution of symbiotic organisms, in some cases, comes along with recombination. For instance, the study of aphid endosymbiont *Arsenophonus* sp. revealed multiple recombination events, even when examining three genes only, indicating extensive genetic exchanges [39]. Even in obligate symbionts, *Blochmannia* and *Buchnera,* indirect evidence of recombination was found; however, they probably refer to ancestral exchanges, enabling lineage adaptation for particular hosts [20,21]. It is considered that recombination is one of the primary mechanisms of genomic evolution for *Bradyrhizobium* sp. and *Sinorhizobium* sp., as it predominantly affects symbiotic regions [40,41,42]. Recombination modulated genetic differentiation in the shipworm endosymbiont *Teredinibacter turnerae* [43]. Finally, *Wolbachia* sp., an endosymbiont of insects, is the most well-studied species in terms of recombination, and has been reported as exerting an effect on both housekeeping genes and loci encoding surface proteins performing contact with host cells [44,45,46]. There is a noticeable pattern in symbiosis-related recombination: the more stable and longer the host-symbiont relationships are, the lower the recombination intensity is. Conversely, an intensive genetic exchange appears to be a characteristic of recently adapted symbionts or those with a wide range of affected hosts.

### 2.3. Pathogenesis

It is not surprising that the overwhelming majority of reported inferences associated with recombination in genomes belong to pathogenic bacteria due to their importance for public health, because it provokes the origin of aggressive isolates causing infectious outbreaks involving *Legionella pneumophila* [47], enteropathogenic *Escherichia coli* O104:H4 [48], *Listeria monocytogenes* [49], and *Neisseria meningitidis* [10,11]. Notably, in some *N. meningitides* strains, a conspicuous HR rate was triggered by horizontal gene transfer [50]. Recombination was demonstrated orchestrating the evolution and genetic differentiation of multiple pathogenic strains relating to diverse bacteria, namely, *Leptospira* sp. [51], *Orientia tsutsugamushi* [52], *Streptococcus pneumonia* [53], *Vibrio vulnificus* [54], *Yersinia pseudotuberculosis* [55], *Pseudomonas aeruginosa* [56], *Vibrio cholerae* [57], *Campylobacter jejuni* [58], *Treponema* sp. [59], and *Ochrobactrum intermedium* [60].

Recombination mediates host adaptation during *Streptococcus dysgalactiae*’s evolutionary history [61]; furthermore, even a single event could lead to alterations in antigenic properties, leading to novel invasive strains [2]. Gene transfer has sufficiently contributed to the evolution of *Salmonella enterica* subspecies, allowing them to infect warm-blooded vertebrates [62]. It has also made the course of infection by serovars Paratyphi A and Typhi remarkably similar, even though they had been initially evolutionarily unrelated [63,64]. Clade divergence in *Chlamydia trachomatis*, corresponding to different infected tissues, has been accompanied by changes in the recombination level [65]. Recombination was also shown to be a major determinant of population structure during the onset of *Clostridium perfringens* infection [66]. Finally, intensive recombination guided the evolution of intracellular parasites *Rickettsia* sp. [67], but the frequency of the exchanges is lowered in more specialized lineages [68]. In some *Helicobacter pylori* strains, the recombination rate exceeded mutations by almost 100 times [69]. Usually, new strains tend to occur during mixed infections, increasing the probability of recombination [70] and concomitant genome rearrangements [71].

Recombination ramifications were also studied in animal and plant pathogens, in which genetic exchange forms new strains, as was reported for *Bartonella* sp., which infects voles [72], bats [73], and even humans [74]. Similarly, recombination governed strain divergence in the salmon pathogen *Flavobacterium psychrophilum* [75], the septicemic agent in pigs *Streptococcus suis* [76], and cattle pathogen *Ehrlichia ruminantium* [77,78]. Finally, recombination induces adaptation to new hosts and virulence changes in plant pathogens, leading to severe economic loss. For instance, recombination represents a genome plasticity determinant in *Xanthomonas citri* [79], *X. gardneri* [80], *X. perforans* [81,82], and *Xylella fastidiosa* [83,84], which infect citrus fruits, tomato, pepper, and coffee species, respectively. Another important example of host adaptation through recombination involves bacteria of the *B. thuringiensis* species. Recombination within the genes coding Cry toxins might play an important role in their diversification and adaptation to new species of insect hosts [85,86,87].

## 3. The Distribution of Recombination Events among the Bacterial Genome

Although virtually all parts of the bacterial genome, including conservative housekeeping genes, are more or less subjected to recombination, there are usually hotspots involved in virulence or antibiotic resistance. A bacterial genome typically contains one or several chromosomes and extrachromosomal plasmids. Recombination can affect both, as they can house mobile genetic elements (insertions, integrative elements, genomic islands) that are transferred during HGT, further causing HR due to the acquisition of homologous regions with high similarity. In the following section, we primarily discuss the role of mobile genetic elements or tandem repeats distributed among the genome, whereas the formation of new alleles and chimeric sequences in the context of certain genes is described in the next section.

### 3.1. Plasmids

Plasmids are extrachromosomal self-replicating genetic elements, the acquisition of which during HGT can induce antibiotic resistance or modulate virulence. Genes located on plasmids are frequently engaged in HR. This has been detected in *Sinorhizobium meliloti* pSymB and pSymA plasmids carrying symbiotic genes [88], the *Staphylococcus aureus* pWBG731 plasmid associated with methicillin resistance [89], the megaplasmid of MDR (multidrug-resistance) *Pseudomonas aeruginosa* strains [90], the IncF plasmid of *E. coli* carrying beta-lactamase [91], and the cp32/cp18 plasmids of the Lyme disease-causing agent *Borrelia* sp. which encode surface-located virulence factors [92,93,94]. Notably, strain B31 of *Borrelia burgdorferi* possesses 12 linear and 9 circular plasmids that exhibit conspicuous evidence of recombination, while the virulence potential descends both from combinations of these plasmids and the recombination within them [95].

Recombination can give birth to novel plasmid types, as shown for the virulence plasmid pAV2 of *Acinetobacter baumanni* [6]; in addition, chimeric plasmids with amino acid synthesis genes were found in symbiotic *Buchnera aphidicola* [20]. Hybrid nature is a typical trait of *B. thuringiensis* toxin-bearing plasmids, such as the camelysin-coding pBMB165 plasmid; pIS56-63 and pBMB0228 Cry toxins-encoding plasmids [96,97]; pAP258 and pAO254 with genes encoding for Vip and Cry toxins [98]. Occasionally, mobile genetic elements can be inserted into a plasmid, which was shown for the vancomycin-resistant *Enterococcus faecium* phenotype possessing Tn1546 transposons on their plasmids [99,100]. Interestingly, in *Klebsiella pneumonia,* plasmid-located insertion sequences caused the incorporation of beta-lactamase genes in the chromosomal genome with further excision of the insertions by homologous recombination [101]. The enterohemorrhagic *E. coli* O26:H-EHEC strain contains a complex plasmid pO26-CRL_115_, simultaneously carrying class 1 atypical integron and two transposons, Tn6026 and Tn21, and each element of the plasmid took part in extensive recombination [102]. Finally, the virulence plasmid pSDVu of *Salmonella enterica* serovar Dublin is considered a source for shortened virulence plasmids belonging to other *S. enterica* strains, while gene elimination during plasmid evolution was associated with recombination [103,104].

### 3.2. Insertion Sequences

Insertions and transposons belong to small mobile elements bearing the transposase gene and are capable of site-specific recombination. Apart from carrying out HGT itself, insertion sequences (IS) often initiate homologous recombination after integrating into a genome, as was demonstrated in the IS-flanked MDR region of *Corynebacterium striatum* [9] or the virulence factors-coding loci flanked by IS1126 and IS1272 of *Porphyromonas gingivalis* and *Staphylococcus haemolyticus*, respectively [105,106]. Insertion acquisition is linked with the emergence of new strains in *Enterococcus faecium* [107] and *Klebsiella pneumonia* [108], and in the latter, further homologous recombination promoted diversification [109]. In *Clostridium botulinum,* neurotoxin-encoding genes are associated with insertions, which mediate inter-strain recombination [110]. Occasionally, insertion-driven recombination may lead to large chromosomal inversions, as for *E. coli* O157 isolate [111], or other genome rearrangements, like gene loss and prophage elimination detected in *Burkholderia mallei* [112].

### 3.3. Long Genomic Regions

More often, yet not exclusively, recombination affects bacterial chromosomes in a gene-scale manner, although it could have an impact on large genome regions, almost inevitably producing new aggressive strains [113]. The consequences of such instances are illustrated by the examples of *Vibrio parahaemolyticus* outbreaks that occurred after large genome rearrangements around O- and K-antigenic regions [113], hypervirulent strains of *K. pneumonia* that underwent 100 kb-long exchanges [114], or strains of *Legionella pneumophila* and *Streptococcus agalactiae*, subjected to similar events throughout their evolutionary history [115,116]. In *Staphylococcus aureus,* such recombination-induced rearrangements cause not only hybrid strain formation [117], but also the extension of infectious potential by gaining the ability to colonize ruminants [118].

### 3.4. Repeats

In some microorganisms, genetic plasticity maintained by recombination is connected to short genome repeats. Tandem repeat Bams30-mediated recombination led to alterations in the exosporium composition and structure in *Bacillus anthracis,* and similar genomic changes contributed to anthrax-like symptoms in several *B. cereus* strains [119]. Bacteria with an extremely compact genome can exploit repeats as a central force for genetic evolution, like *Mycoplasma pneumonia,* which possesses one of the smallest genomes, yet exhibits recombination signals around repeats [120].

### 3.5. Genomic Islands and Integrative Elements

Genomic islands are horizontally acquired loci of varying lengths embedded into the chromosome. In a broad sense, this term describes staphylococcal chromosomal cassettes and integrative and conjugative elements (ICEs) that exploit the bacterial conjugative apparatus for self-replication [121]. Similar to insertions, these elements often flank antibiotic resistance or virulence genes and promote recombination, as was observed in *Vibrio cholera* ICEVchInd5 [26], *Streptococcus pneumonia* Tn5253-containing ICE [122], and *Staphylococcus sciuri* SCCmec III (staphylococcal cassette chromosome element) [123]. Special AICEs (actinomycete integrative and conjugative elements) in *Streptomyces* sp. ensure diversity of antimicrobial compounds through active genetic exchanges [38].

Pathogenicity islands represent one of the main targets for modern clinical microbiology because their transmittance and acquisition by bacteria induce the development of new pathogens. Moreover, the acquisition of these islands is correlated with an increased recombination rate, thus providing adaptation to hosts. Pathogenicity islands with concomitant traces of recombination were identified in the plant pathogens *Acidovorax avenae* [28] and *Pseudomonas viridiflava* [124] and numerous human infectious agents and opportunistic pathogens, including *Streptococcus suis* [125], *Clostridioides difficile* [126], *Actinobacillus actinomycetemcomitans* [127], *E. coli* [128], and *Pseudomonas aeruginosa* [129]. Moreover, symbiotic islands (SI) can encode symbiotic genes required for making contact with the host, and genes within them are prone to adaptive recombination, as in the case of *Bradyrhizobium* sp. [41].

### 3.6. Prophages

Prophages often serve as a means of horizontal gene transfer, and their acquisition is associated with subsequent recombination, which is commonly associated with antibiotic resistance such as in *Clostridioides difficile* [27] and *Salmonella enterica* var *Typhimurium* [130], or virulence as in *Streptococcus pneumoniae* [131]. Finally, *Bartonella* sp. bears demystified bacteriophages called GTAs (Gene transfer agents), the assembly of which is controlled by the host; thus, bacteria can utilize GTAs to amplify genes essential for infection [132].

## 4. Functional Characteristics of Genes Subjected to Recombination

### 4.1. Surface Proteins and Adhesion Factors

Membrane proteins are required to establish contact with host cells, being crucial for parasitic and symbiotic relationships at the onset and/or after establishment, biofilm formation, and antigenic properties. The diversity of these proteins accounts for the varieties of both virulence and symbiotic potential. Evidence of recombination is detected in *Wolbachia* sp. genes encoding surface proteins, namely Wsp [133] and ankyrin-rich (ANK) [45], in particular, cidA and cidB, which control cytoplasmic incompatibility [46]. Genetic diversification of *Borrelia* sp. with diverse infection strength is linked to recombination-mediated variations in membrane proteins OspA и OspB [92], surface antigen EppA [134], and OspE/F-like lipoproteins [135]. Similar hypervariability patterns are characteristics of OmpA and pmpE/F/H in *Chlamydia trachomatis* [136,137,138], incA in *Chlamydia pneumonia* [139], *Mycoplasma pneumoniae* P1 adhesin [120], *Helicobacter pylori* babA and babB proteins [140], *Listeria monocytogenes* internalins [141], *Moraxella catarrhalis* UspA1 and UspA2 proteins [142], pilus proteins and pspA in *Streptococcus pneumoniae* [143,144], *Acinetobacter baumanni* adhesins [145], Opa in *Neisseria meningitidis* [146] and membrane proteins in *Legionella pneumophila* [147], *Staphylococcus aureus* [148], and *Streptococcus agalactiae* [149].

Genes encoding surface proteins often have a mosaic structure, which assists in evading host immunity. This recombination-emanated mosaicism was found in fibrillar anti-phagocytic *Streptococcus pyogenes* proteins [150], immunoglobulin-like ligA, ligB, ligC of *Leptospira* sp. [151], autolysin and spA of *Staphylococcus aureus* [152,153], and surface lipoproteins fHbp of *Neisseria meningitides* [154]. In addition, in *N. meningitides,* an intensified recombination rate was induced shortly after the horizontal acquisition of the *pgl* locus, regulating glycosylation patterns; thus, rapid antigenic switches were observed [50]. In *Treponema* sp., the variety of antigens is delineated by the repeat-associated recombination in the *tpr* gene family [155].

### 4.2. Secretion Systems

Bacterial secretion systems, comprehensive complexes of membrane proteins, carry out the secretion of infectious effectors during the infection course. Therefore, it is no wonder that allelic diversity in the respective loci modulates virulence potential. Most of all, genes encoding type III secretion systems in *Chlamydia trachomatis* [65] and plant pathogens *Pseudomonas syringae* [156] and *P. viridiflava* [124] have been affected by recombination.

### 4.3. Infection Effectors

Effectors are typically represented by secreted proteins that control specific infectious stages, especially for intracellular pathogens. Similar to toxins, allelic diversity in respective genes may promote adaptation to particular hosts. Recombination has affected the sidJ and DotA effectors, which provide vesicle stability of *Legionella pneumophila* inside host phagocytes [157,158], transferring receptors of *Mannheimia haemolytica* [159] and *Neisseria meningitidis* [160], which modulate the infection course via ferric ion influx, and secretive glycosyltransferases of *Streptococcus salivarius* that produce a matrix for dental plaques [161].

### 4.4. Toxins

Allelic diversity in toxin-coding genes entails alterations in virulence or the course of infection. Thus, it is likely to be seen in genes encoding toxic compounds in multiple human pathogens, including *Clostridium botulinum* neurotoxins [162,163], *Clostridioides difficile* TcdA and TcdB (toxins A and B) [164], *Mannheimia haemolytica* leucotoxins [165], *Streptococcus pyogenes* streptolysin [166], *E. coli* enterotoxins [167], and *Helicobacter pylori* CagA oncoprotein [168]. In *B. thuringiensis,* insecticidal potential is maintained either by plasmid recombination, dovetailing diverse combinations of insecticidal moieties in different strains, or probable intra-gene recombination, leading to domain swapping between Cry toxins [85,86,87].

### 4.5. Antibiotic Resistance Genes

The development of antibiotic resistance is usually caused by the acquisition of mobile genetic elements associated with resistance genes. Obtained loci tend to recombine and mutate more frequently, hampering the prolonged application of antibiotics within one chemical class. Notably, some organisms, such as *Acinetobacter baumanni,* can become a reservoir of resistance for other bacteria, e.g., enterobacteria [169]. Events related to plasmids, ICEs, and insertions were reviewed above; nevertheless, recombination is found in single chromosomal genes. Examples include loci encoding permeases and transcriptional factors linked with tigecycline insensitivity in *Acinetobacter baumannii* [170], genes ensuring resistance to beta-lactams in *Burkholderia multivorans* [171], and housekeeping genes *gyrA* and *parC* associated with fluoroquinolone resistance in *E.coli* ST1193 clone [172].

### 4.6. Polysaccharide Synthesis

Outer membrane lipopolysaccharides and capsular polysaccharides act as prominent antigens for immune cells and vaccines; thus, recombination-induced variability causing serotype switching is considered to be a crucial problem in medicine, requiring the development of novel approaches in vaccine production. Recombination hotspots were located in loci controlling polysaccharide and lipopolysaccharide biosynthesis in *Enterococcus faecium* [173], *E. coli*. [174], *Klebsiella pneumoniae* [174], *Helicobacter pylori* [175], and *Legionella pneumophila* [147]. Importantly, these events lead to the formation of so-called clinical strains, causing nosocomial infection [173]. Recombination-maintained capsular switching, responsible for immune evasion, was observed in many pathogens, including *Acinetobacter baumanni* [176], *Neisseria meningitidis* [177], *K. pneumoniae* [174], *Streptococcus agalactiae* [149,178,179], *S. pneumoniae* [180], *S. pyogenes* [166], and the *E. coli* ST131 strain [181].

### 4.7. Metabolic Pathways

Recombination in loci encoding components of the metabolic pathways can enhance adaptation by acquiring the ability to catabolize new energy sources or by modulating the efficacy and speed of the involved enzymes. For example, recombination caused duplication and subsequent diversification of naphthalene metabolic gene clusters in *Pseudomonas stutzeri* [13]. Signals of homologous recombination were also revealed in the cellobiohydrolase gene in *Teredinibacter turnerae* [43], or gene clusters for nicotine and carbohydrates catabolism located on the horizontally acquired pAO1 megaplasmid in *Arthrobacter nicotinovorans* [35]. These respective genetic changes provided an adaptive advantage via expanding the metabolic potential in soil and host microenvironments. Finally, recombination affected the polyketide biosynthesis gene cluster in *Streptomyces* sp., thus enabling it to compete more efficiently with other bacteria by synthesizing novel antimicrobials [182]. Similar recombination-driven alterations were found in genes encoding signaling Nod-factors synthesis enzymes in *Sinorhizobium* sp. regulating molecular interactions with novel hosts when establishing symbiotic relationships [42].

## 5. Conclusions

Recombination-driven gene transfer directs prokaryotic evolution as a fundamental mechanism of genetic exchange. Having reviewed current genomic studies, we found that recombination hotspots tend to be located near horizontally acquired genes, mobile elements, and genome repeats (Figure 1a). These respective events could be classified according to three major trends, namely (i) ecological diversification, (ii) pathogenesis, and (iii) symbiosis. These exchanges mostly affect genes encoding membrane proteins, toxins, antibiotic resistance factors, polysaccharide biosynthesis enzymes, effectors, secretion systems, and metabolic pathways, with only the latter being involved in all three evolutionary trends mentioned (Figure 1b). Presumably, when microorganisms occupy a new environment, genetic exchanges between them and concomitant rearrangements occur more frequently. For instance, recombination occurs at the initial stage of adaptation to new hosts or denotes the continuous process of diversification, as in the case of Cry toxins in *B. thuringiensis*. Nevertheless, over time, relationships with the host become increasingly specialized, accompanied by an incremental decrease in recombination rate. Therefore, the signals of recombination are primarily detected in loci associated with environmental interaction and can be used as genomic markers displaying the ecological history of the studied strains.

In the reviewed studies, predominant methods for detecting recombination events were based on computational predictions in the genomic data. However, despite a large arsenal of tools available, current methods are not free from limitations, e.g., dependence on the models and dubious assumptions that the core genome reflects clonal relationships [5]. Nevertheless, even with possible artefactual inferences, the data from current genomic studies provided similar functional groups of genes to those reported in laboratory studies. This indicates that computational pipelines seem to correctly display the evolutionary dynamics of bacterial genomes in the context of recombination. It is noteworthy that, when analyzing the functional role of genetic exchanges, we did not select reports with predetermined criteria of grouping, however, we did reveal general trends. Undoubtedly, there is a skew toward pathogenic bacteria due to threats to global health; thus, there is a need for further studies on species that occupy other ecological niches.

Observed genomic regions with increased recombination rates might serve as a roadmap for further studies. Possible implications could include targeted analysis of genomic loci housed in mobile genetic elements, such as genomic islands or those flanked by insertions. Another strategy lies in picking particular genes, e.g., toxins of secretion systems with the reconstruction of recombination events, both ancestral and recent. Subsequent matching of these inferences with species’ phylogeny would help to reveal key adaptation steps to novel environments and identify common and/or distinct pathways within different ecological groups. All things considered, there is a strong demand for (i) performing comparative studies of recombination intensity across different bacterial species, (ii) developing new mathematical models and novel bioinformatic tools for recombination detection, and (iii) carrying out experimental validation of computationally derived observations to yield new insights and deepen our understanding of an intricate network of recombination events with their functional ramifications.

## Figures and Tables

**Figure 1 toxins-15-00568-f001:**
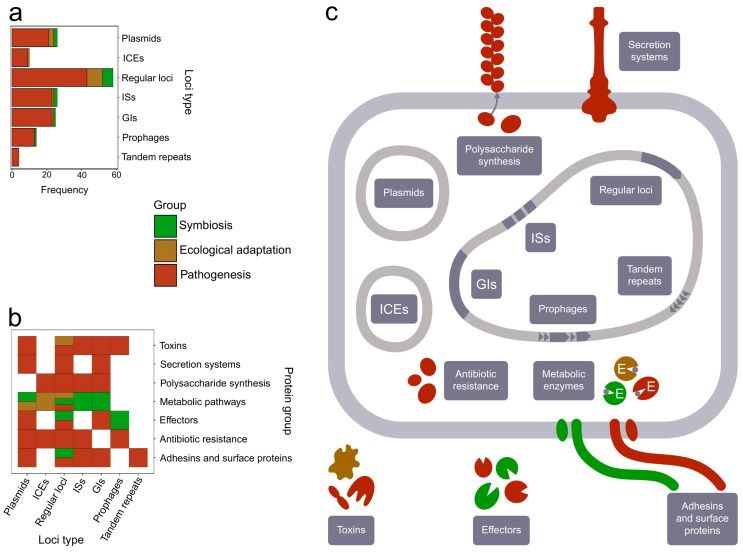
Functional impact of loci subjected to recombination and their genomic context. In each case, the color corresponds to the consequences of recombination events, classified into three main groups, namely, ecological adaptation, symbiosis, and pathogenesis. The list of the analyzed events from 91 bacterial species is presented in Appendix A. Used abbreviations are as follows: ICEs—integrative and conjugative elements; ISs—insertion sequences; Gis—genomic islands. (**a**) The absolute group-wise number of events attributed to different types of genomic regions. Regular loci imply genes and/or operons that are not surrounded by or located within mobile genetic elements. (**b**) The distribution of loci types encoding protein products attributed to their functional roles. Each tile implies that a certain protein group is encoded by a particular genomic locus. (**c**) A schematic pictorial representation of the structures and processes in which proteins encoded by loci subjected to recombination take part.

## Data Availability

Not applicable.

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
