# Peer review of "Recombination in Bacterial Genomes: Evolutionary Trends"

_toxins, 2023, doi:10.3390/toxins15090568_

Round 1
Reviewer 1 Report
This review paper provided a detailed summary of the role and impact of genetic exchanges including homologous recombination (HR) and horizontal gene transfer (HGT) in bacterial evolution and adaptation. Expertly organized, it not only surveys a wide range of studies to bolster its claims but also emphasizes the critical roles of homologous recombination and horizontal gene transfer in bacterial diversity, speciation, pathogenesis, and ecological interactions in the bacterial world. Thereby, highlighting the need for more focused research and practical applications surrounding recombination. The review also reveals that the frequency of these genetic exchanges is often influenced by the bacteria's ecological niches, offering key insights that have significant implications for antibiotic resistance and vaccine development. Overall, this review provides comprehensive insights into bacterial genomics and the mechanisms of its evolution. It merits publication.
Author Response
The authors would like to thank the reviewer for the high assessment of our work and for highlighting its significance to the field.
Reviewer 2 Report
toxins-2575126
This is an evolutionary review that summarizes genetic exchanges and recombination hotspots in bacteria. They found genes coding for toxins among others. The review is certainly comprehensive and well written. The Reference section is lengthy and appropriate to the topic.
Minor comments:
1) Abstract. I was waiting for several bacterial species. Please, clarify in the abstract which bacteria you are talking about.
2) Reading the text it becomes clear that the review encompasses a large number of bacterial species. Still, I like to know what the selection criteria were to include certain bacteria and not others.
3) Many of the functional characteristics of the genes subjected to recombination were previously known (e.g. adhesion factors, secretion systems, toxins, antibiotic resistance genes. Please, delineate what is new about your conclusions?
Author Response
We are grateful to the reviewer for their positive assessment of our work and for raising crucial issues to be addressed. Below please find our point-to-point answers to the comments.
- We have refined the abstract so that it is now focused more on the aim of the review. We also explicitly stated the number of bacterial species used in our analysis both in the abstract and the Introduction section.
- We agree with the reviewer that it was unclear which criteria were applied to pick publications. In our work, we aimed to reveal some common trends in the context of functional ramifications of recombination events in 91 bacterial species. The comprehensive table with all necessary information, including species names, subjected loci, and functional impact of the exchanges/acquisitions is provided in the Supplementary material as Tabe S1. We deliberately moved the source table to Supplementary given its high volume. We tried to refer interested readers to the table in the text. The text itself was concise to highlight only the most important representative cases, while overall information is displayed in Figure 1a-b. As for criteria of inclusion, we searched for the articles that meet the following criteria: (i) the research should be relatively recent; (ii) genetic exchanges, either homologous recombination and/or horizontal gene transfer, are detected computationally; (iii) it is possible to reveal functional ramifications of the events. For instance, if the species were analyzed by simple MLST schemes, and no chimeric loci and/or particular genomic regions were provided, such articles were not considered. As a result, most of the studies included represented analysis of the whole genomes.
- Indeed, most of the inferences represented known examples of genes subjected to recombination more often than others. Nevertheless, we should note that the primary aim of the review was to perform some general classification but not reveal novel instances. We find that the most important conclusion is the presence of three major trends regarding ecological groups of the bacterial species. Such grouping was obtained by bottom-up but not top-down approach. We first reviewed the articles and then noticed that all of the instances fall into three trends fitting well with the functional categories of the proteins encoded by recombination-affected loci. Moreover, the frequency of certain recombination hotspots juxtaposed with both ecological trends and the biological role of the products encoded by the respective loci might provide insights into which molecular hallmarks of adaptation might be common for diverse ecological groups of bacteria (e.g., metabolic pathways). Another observation made is a source of grey areas in the field of bacterial genomics is demonstrated by Figure 1b. One can observe that most combinations of loci types bearing genes related to certain functional categories appear to fall into a particular ecological category which is apparently caused by a skew towards genomic studies of pathogenic bacteria. We revised the Conclusions section accordingly to emphasize the key findings of our review and proposed future directions. Current reviews describe certain bacterial species, particular ecological groups, or the general role of recombination. However, to the best of our knowledge, there are no summaries aimed at classifying the events irrespective of a predetermined selection to reveal the frequency of particular types of genetic rearrangements. We, therefore, hope, that our review will be useful for researchers in the field.
Reviewer 3 Report
The manuscript ID toxins-2575126 compiles information about bacterial genomes' complex recombination-driven evolutionary routes for adaptive purposes to the environment. The review has interesting elements and relevant information for readers. However, some issues should be addressed prior to further consideration.
1. This review seems to be a mini-review. The information is highly condensed, and more examples and cases should be added to the manuscript to expand the information for readers.
2. The abstract should be revised since it is too large, and the aim and scope of this review are not clearly defined. Several ideas are embodied, justifying some introductory facts, but lack clarity about aim and scope. In addition, a conclusive sentence should be added to the abstract ending to define an outlook regarding this review. Be consistent throughout the manuscript.
3. A scheme outlining the three topics of the functional impact of recombination (section 2) should be added to the manuscript.
4. A table summarizing the examples of the functional characteristics of genes subjected to recombination (section 4) must be provided for readers.
5. A description of the gene recombination and/ or transfer of biosynthetic gene clusters for producing specialized metabolites with ecological roles (e.g., antibiotics or signaling molecules) is missing in section 4.
6. A section dedicated to experimental and bioinformatics tools and methods for recombination detection and analysis in bacteria can be added to improve the review content.
7. A future outlook and perspectives section is missing.
8. The conclusions section must be revised since it comprises a summary of gathered information involving some introductory passages. However, no conceptual findings are provided after analyzing the compiled data and sources.
Detailed scrutiny should be performed throughout the manuscript to look for a few grammar and stylistic issues.
Author Response
We thank the reviewer for a thorough analysis of the manuscript. The provided suggestions and critical comments were valuable for improving our review.
- Forasmuch as the aim of the review was the characterization and categorization of genetic exchanges driven by homologous recombination and HGT on the basis of current genomic studies, the review is concise to avoid repetitive information. We included recent studies from 91 bacterial species of different taxonomical categories whose genomes showed signals of recombination events which, as we believe, is sufficient to perform classification. The detailed table with the source data from 192 articles is included in the Supplementary material in Table S1. We emphasized it in the text for interested readers. We did not aim to perform a detailed description of recombination processes for certain species but rather tried to assess differences and common features of genetic exchanges in bacterial species with different ecological niches in the context of the types of subjected genomic regions and functional roles of the products encoded by loci within these regions. Figures 1a-b provide a brief illustration of the gathered data enabling us to find similarities and differences within three ecological categories and identifying poorly studied areas in the field.
- As suggested, we reduced the abstract by omitting introductory passages. We also clarified the scope of the article, i.e., revealing common trends in ramifications of recombination events found in bacterial genomes. We also added the concluding sentence.
- We believe that Figure 1 covers the information from all three sections and illustrates both types of suggested loci, functional groups of proteins encoded by them in the context of the ecological group which is color-coded.
- The respective table (Table S1) is provided in the Supplementary material. It summarized species, genomic loci, and functional impact of the events. Given the size of the table, we decided to move it to the Supplementary material, however referring the reader to the table throughout the text.
- As required, we have expanded section 4.7 and provided the requested details on the roles of metabolites encoded by loci subjected to recombination
- Recently, we have published a detailed review of cutting-edge bioinformatic tools for recombination detection outlining the strengths and limitations of certain methods. On that account, including such a section in the current manuscript would simply repeat the mentioned information. Nevertheless, we briefly discussed the problems of computational predictions in the Conclusions section.
- Given that we discarded introductory information from conclusions, the perspective of future research was added to the respective section.
- The Conclusions section was revised according to the reviewers’ suggestions. We reduced common information and highlighted key considerations. The text now is consistent with the aim. We find that the most important observation is the fact that without cherry-picking particular species, the reviewed events could be classified into three major evolutionary trends. We have searched for articles reporting recombination and/or gene transfer in bacterial genomes. To the best of our knowledge, there is a host of reviewers dedicated to molecular mechanisms of recombination and its role in the evolution of particular species, whereas overall summaries that provide the frequencies of the events in the context of their functional role and genomic context irrespective of species are absent. We also discussed a possible roadmap for future studies based on our observations.
As requested, we have checked the manuscript, corrected spelling errors, and addressed several stylistic issues.
Round 2
Reviewer 3 Report
The authors adequately addressed my comments so that the manuscript can be accepted in its current form.